# Natural Compound Boldine Lessens Myotonic Dystrophy Type 1 Phenotypes in DM1 Drosophila Models, Patient-Derived Cell Lines, and HSA^LR^ Mice

**DOI:** 10.3390/ijms24129820

**Published:** 2023-06-06

**Authors:** Mari Carmen Álvarez-Abril, Irma García-Alcover, Jordi Colonques-Bellmunt, Raquel Garijo, Manuel Pérez-Alonso, Rubén Artero, Arturo López-Castel

**Affiliations:** 1Valentia BioPharma S.L., 46980 Paterna, Spainruben.artero@uv.es (R.A.); 2Human Translational Genomics Group, Institute for Biotechnology and Biomedicine (BIOTECMED), University of Valencia, 46100 Burjasot, Spain; 3Incliva Biomedical Research Institute, 46010 Valencia, Spain

**Keywords:** boldine, myotonic dystrophy, rare disease, Drosophila, drug development, natural small molecule, patient-derived cells

## Abstract

Myotonic dystrophy type 1 (DM1) is a complex rare disorder characterized by progressive muscle dysfunction, involving weakness, myotonia, and wasting, but also exhibiting additional clinical signs in multiple organs and systems. Central dysregulation, caused by an expansion of a CTG trinucleotide repeat in the DMPK gene’s 3’ UTR, has led to exploring various therapeutic approaches in recent years, a few of which are currently under clinical trial. However, no effective disease-modifying treatments are available yet. In this study, we demonstrate that treatments with boldine, a natural alkaloid identified in a large-scale Drosophila-based pharmacological screening, was able to modify disease phenotypes in several DM1 models. The most significant effects include consistent reduction in nuclear RNA foci, a dynamic molecular hallmark of the disease, and noteworthy anti-myotonic activity. These results position boldine as an attractive new candidate for therapy development in DM1.

## 1. Introduction

Myotonic dystrophy type 1 (DM1) is the most common adult-onset muscular dystrophy, characterized by severe muscular defects, but also exhibits a puzzling time of onset, and a wide spectrum of comorbid clinical presentations, from cognitive impairment to endocrine abnormalities [1]. Traditionally, the overall prevalence has been estimated at around 1 in 8000–10,000 individuals. However, recent population-wide screening suggests that the actual disease burden might be much higher than previously thought. In fact, a mutation prevalence of 4.8 in 10,000 individuals has been found in some populations, making DM1 one of the most common rare diseases [2,3]. The key muscular features of DM1 include myotonia, muscle weakness, and wasting, but 3manifestations of cardiac conduction defects and respiratory insufficiency, the two predominant causes of mortality, are also important. Specifically, myotonia is one of the earlier and most prominent DM1 symptoms manifesting as the delayed relaxation of muscles after voluntary contraction or percussion. Myotonia contributes to several debilitating phenotypes such as decreased dexterity, gait instability, difficulty with speech/swallowing, and muscle pain [4]. 

DM1 is caused by pathogenic expansion of an unstable CTG microsatellite in the 3′ UTR of the DM1 protein kinase (*DMPK*) gene, transcribed to an RNA messenger with an expanded CUG (CUG_exp_), leading to the formation of ribonuclear foci in the nuclei of affected cells, the most prominent histopathological hallmark of the disease. [5]. These foci interfere with proteins that regulate alternative splicing (AS) and contribute to DM1 phenotypes by inducing fetal AS patterns in adults. The loss of Muscleblind-like (MBNL) factor function, the most recognized cause of disease pathogenicity development, accounts for more than 80% of missplicing events in patient samples. This occurs through well-established CUG_exp_ sequestration, but in recent studies it is documented as prompted by specific microRNAs-based translational repression of MBNL1 and 2 transcripts isoforms [6,7].

Several treatment approaches for DM1 are currently under development, among which some candidates are already in clinical phases (recently reviewed in [8]), but as of yet, none have been approved for clinical practice, highlighting an urgent need to discover valid drugs. Repurposed drugs, such as metformin, mexiletine, erythromycin and tideglusib [9,10,11,12], together with natural molecules [13,14], are closest to market approval [8]. These therapies have the potential not only to improve neuromuscular defects, but in some cases combat the multisystem and degenerative aspects of the disease. Boldine is a well-characterized alkaloid natural molecule, still not approved as a drug, but with a large number of health-promoting properties already described, including targeted muscle effects [15,16,17,18]. Identified through previous high-throughput screening based on the use of a transgenic Drosophila melanogaster model [19], here, we describe the validation and characterization of its potential use linked to rescuing DM1-linked phenotypes. 

## 2. Results

### 2.1. Boldine Improves DM1 Phenotypes in Flies

Previously, we developed an in-house fully automated Drosophila-based platform that enabled chemical traces to undergo in vivo high-throughput screening (HTS) in 96-well plates. We used spliceosensor flies for reliable AS quantification of human INSR (*insulin receptor*) minigene under the absence (MHC-Gal4>UAS-INSR:Luc#6 flies. No-DM1 state) or presence of pathogenic CTG expression (MHC-Gal4>UAS-INSR:Luc#6; UAS-iCTG480 flies DM1 state) [19]. Specifically, we studied the ability of individual compounds to change luminescence levels (luciferase quantification) on spliceosensor fly homogenates (three larvae/well) in DM1 condition. In the initial trial, 126 of the 16,063 compounds tested were identified as positive hits (transition to a non-DM1 state, (Appendix A)), as previously described [19]. Looking for existing similar chemical structures among the hits, we detected two phenanthrene-based molecules. Phenanthrene is a polycyclic aromatic hydrocarbon (PAH) with the formula C14H10, consisting of three fused benzene rings (Figure 1a). This structure is common in approved drugs, particularly alkaloids such as morphine and codeine (Figure 1a) [20]. Based on the above, we selected both similar hits for validation studies. Here, results are shown for one of them, boldine, the most dominant alkaloid from the almost 20 distinct alkaloids comprising the plant *Peumus boldus* [21] (Figure 1b).

During the primary screening, luciferase levels were normalized by dividing the total luciferase values by the number of flies in the well. However, non-specific increases of luciferase levels in both the indirect musculature and the entire body of the fly could have led to false positives. To validate boldine activity, we performed an experiment to analyze individual luciferase levels from at least eight individual spliceosensor flies (eight points test), DM1 state treated with the same concentration used in the primary screening (Appendix A) [19]. As a control, we used the same flies fed with dimethyl sulfoxide (DMSO), the common solvent for most of the compounds during the primary screen. The luminescence levels in individual flies treated with boldine were significantly higher than those of the flies fed only with DMSO. These results confirmed the activity of boldine as reproducible and reduced the possibility of a false screening positive. 

One additional potential concern with the primary screening results is that boldine might display off-target effects. This could happen either over the luciferase reporter signal or affecting the UAS/Gal4 system that directs the muscle expression of the CTG repeats and the spliceosensor minigene, and leading to non-specific luciferase increased levels. To discard this possibility, we first treated flies expressing luciferase, but without the expression of the spliceosensor minigene or the CUG repeats (MHC Gal4; UAS-Luc) with boldine or DMSO. We found that luminescence levels were similar in both groups (Appendix A). This result indicated that boldine does not directly affect luciferase expression, processing, or stability. Secondly, we measured the luciferase activity of spliceosensor flies, that involves the expression of the *INSR* minigene, but no expression of the CUG repeats, after treatment with boldine or DMSO. We observed similar luminescence levels in both groups, confirming that boldine also does not affect the luciferase expression of the *INSR* minigene in the no-DM1 state (Appendix A). Therefore, the increase in luciferase activity observed in spliceosensor flies after the treatment with the boldine is confirmed to be a disease-specific effect.

One of the main histopathological features of DM1 is the accumulation of foci, which are ribonuclear aggregates formed by toxic double-stranded hairpin RNA and various nuclear proteins frequently involved in AS regulation [5]. Given that boldine was effective in rescuing a muscleblind-dependent splicing event (*INSR*), we investigated if this result was the consequence of its ability to modulate the focal formation (Figure 2a). The results showed that boldine significantly reduced the presence of ribonuclear foci in indirect flight muscles (IFMs) analyzed on longitudinal sections of the thorax of spliceosensor flies in the DM1 condition. The percentage of cells with the foci decreased was 30.2% in the flies treated with boldine, as compared to the 60.1% in those treated only with the solvent (DMSO) (Figure 2a). To further validate the potential benefits of boldine activity in DM1 flies, we conducted a functional assay in the MHC-Gal4>UAS-i(CTG)480 DM1 line (DM1 state) that displays a consistent reduction in its lifetime frame. We monitored the survival of flies for 50 days, with some flies receiving boldine or DMSO. Boldine treatment significantly increased the longevity of DM1 flies compared to the control group receiving DMSO, as evidenced by the statistically significant differences between their survival curves (Figure 2b). The wild-type flies (*yw*) treated with DMSO were used as a positive control for survival. 

The experiments carried out in different models of DM1 in Drosophila have confirmed that, in this organism, boldine improves phenotypes related to human disease at both molecular and phenotypic levels.

### 2.2. Activity of Boldine in DM1 Human Cell Lines

The next goal was to evaluate the compound activity in the human context. To this end, we designed experiments using fibroblasts obtained from either healthy individuals (9.88 cell line) or DM1 patients (9.73, 9.56, and 9.66 cell lines), kindly donated by Dr. Adolfo L. of Munain Hospital of Dosnosti. These fibroblasts were previously stably transduced with MyoD, as described in [22], which enabled their transdifferentiation into myoblasts. Since boldine reduced the presence of ribonuclear foci in Drosophila, we interrogated the same skill in human cells. We tested the compound’s activity on toxic RNA foci in fibroblasts derived from the myoblasts of patient carrying 333 CTG repeats (9.73 cell line). These myoblasts were incubated for 24 h with 100 μM boldine or with 1% DMSO (control). The presence of ribonuclear foci was determined by fluorescent in situ hybridization using a CAG probe (Figure 3a). While, as expected, no foci were detected in the myoblasts derived from healthy individuals (9.88 line), foci were consistently found in DM1 cells. The treatment of 9.73 myoblasts for 24 h reduced the number of foci present in the cell nucleus from a mean of 2.7 foci per control cell (DMSO) to a mean of 1.3 foci in cells with boldine. In addition, boldine treatment increased the number of myoblasts without foci compared to treatment with DMSO. We conclude that boldine is also capable of lessening this typical histopathological alteration associated with DM1 in the human context.

Following on from these results, we investigated whether boldine could also enhance MBNL1distribution and increase the free protein levels (Figure 3b). In the no-DM1 cells, MBNL1 is evenly distributed throughout the nucleus (9.88 line), whereas most of the MBNL1 protein is sequestered by CUG repeats in DM1 cells in the form of ribonuclear foci, as previously characterized [5]. Specifically, we treated DM1 myoblasts (9.56 line) with 100 μM boldine or DMSO (1%) (Figure 3b). Unexpectedly, the boldine treatment did not change the distribution of MBNL1 in DM1 myoblasts, and most of the protein remained grouped in small areas of the nucleus. These results suggest that while boldine reduces the focal formation in the myoblasts of DM1 patients, this ability it is not connected to the recovery of free MBNL1 levels in the nucleus, potentially limiting its further activity on the downstream connected disease features, such as AS.

The transdifferentiated DM1 myoblasts display splicing alterations in several transcripts, such as in exon 22 of the sarcoplasmic/endoplasmic reticulum calcium-dependent ATPase (*SERCA*), excluded at all times, and resembling its fetal pattern even in adults [23]. Another example is the cardiac troponin T (*cTNT*), a protein found in the sarcomeres of fast muscle fibers in skeletal muscle [24]. The splicing pattern of this gene is altered, and the fetal exon 5 is maintained in DM1 patients. Using myoblasts from patients with 1000 CTG repeats (9.66 line) and healthy myoblasts (9.88 line), we analyzed the splicing of *cTNT* and *SERCA* transcripts treated with 100 µM boldine or DMSO (1%) (Appendix A). After analyzing the percentage of inclusion of *cTNT* exon 5 in 9.66 cells, we found that boldine treatment did not modify the disease-linked inclusion of this exon compared to DMSO treatment. The same result for disease-linked exclusion of *SERCA* exon 22 was observed. We replicated the two gene AS evaluation in a cellular context with fewer toxic repeats, using 9.73 DM1 myoblasts carrying 333 repeats. Consistently, boldine treatment did not modulate disease-linked AS events [25]

In summary, boldine treatment reduced focal formation in DM1 fibroblasts, but this is not followed by a correction of the spatial reorganization of MBNL1 or the recovering of disease-linked AS events. Thus, the cells maintained part of their DM1-linked dysregulation. 

### 2.3. Validation of the Boldine Activity in a Murine Model of DM1

For a robust validation of the in vivo activity of boldine, we selected the HSA^LR^ murine model of DM1. This model expresses 250 CUG repeats under the control of human actin promotor and reproduces molecular to functional symptoms of DM1, such as ribonuclear focal formation, alterations in transcript splicing, and myotonia in skeletal muscle [26]. In order to minimize the impact of mouse model variability on the results, mice aged four to six weeks were selected for all the experimental approaches described. First, boldine treatment was carried out via intramuscular injections at two concentrations, 60 mg/kg and 25 mg/kg, for focal quantification (Appendix A). Briefly, HSA^LR^ mice were treated daily with the compound or DMSO for three consecutive days, and then sacrificed two days after the administration of the last dose. In mice treated with boldine at 60 mg/kg, in situ fluorescence hybridization analysis showed a reduction in the average percentage of nuclei with foci in the quadriceps, reaching 15%, versus the percentage observed in the quadriceps treated with DMSO only (49%). This effect was observed in 50% of the treated mice (n = 6) (Appendix A). The same effect was observed after treatment at 25 mg/kg (12% nuclei with foci) compared with DMSO injections (49% nuclei with foci), and the reduction displayed in the 66% of the treated mice (n = 3) (Appendix A). Overall, these results confirmed boldine reproducible ability to significantly reduce the ribonuclear focus DM1 hallmark.

Additionally, we explored the ability of boldine to restore splicing defects in these mice. Exon 22 of the ATPase-dependent calcium Serca (*Serca*) and exon 7a of the type 1 chlorine channel (*Clcn1*) splicing transcripts were chosen for evaluation due to their conserved splicing defect in both DM1 patients and HSA^LR^ mice. For *Clcn1*, exon 7a inclusion leads to the presence of a premature stop codon during messenger processing, resulting in a truncated protein and decreased functional CLCN1, which is related to myotonia in DM1 [26,27]. Using this approach, we also included the no-disease FVB animals (with the same genetic background as the DM1 mice) as controls for AS events for comparison purposes. Consistent with the previously observed cell models, RT-PCR analysis of *Serca* and *Clcn1* transcripts from HSA^LR^ animals treated with boldine and DMSO did not reveal splicing modulation [28]. 

Together, these results display reproducible results for boldine activity on different early molecular and cellular DM1 features in different disease models, with an ability to reduce ribonuclear foci, but not connect to a relocation of a functional MBNL1 or AS modulation. 

Nevertheless, we took advantage of the murine model to also evaluate a functional disease feature. HSA^LR^ mice develop myotonia from four weeks of age [26]. A first approach involved the intramuscular injection of five different boldine concentrations (n = 3 to 7, from 60 mg/kg to 6.25 mg/kg) in the left quadriceps of the hind legs, while the right quadricep was injected with DMSO (20%) (Figure 4a). Electromyograms were performed before the first injection and two days after the last dose, with daily treatment for three days, as previously described. As expected, the high degree of myotonia was the same for both legs before the treatment. The results showed that boldine significantly reduced the local levels of myotonia in a dose-dependent manner in the left quadriceps. The contrary leg of same mice treated with vehicle alone consistently kept initial high values of myotonia. This finding suggests that boldine has the ability to reduce the myotonia levels in DM1 mice. At this point, this result was unexpected, since no correlation with negative AS results were previously observed. However, after the local positive effect, we extended the evaluation to a systemic administration of boldine in the same model. We treated HSA^LR^ mice (n = 3–6) with boldine intraperitoneal (daily treatment for three days at 30 mg/kg) and via an intragastric route (daily treatment for five days at 25 mg/kg). Myotonia levels were analyzed shortly after the administration of boldine via both routes, but no reduction in myotonia levels were observed after daily quantifications for five days after the last dose [29]. We also monitored the need for a larger period of systemic delivery to see the effect of boldine in muscles. We measured myotonia levels for 21 days after the intraperitoneal administration of the molecule at 30 mg/kg. Again, negative results for myotonia reduction were detected [30]. The fact that boldine is only active in reducing myotonia in DM1 mice after intramuscular administration in our experimental approaches suggests that its distribution and pharmacokinetic properties in the whole organism may be limited. 

Given boldine’s ability to reduce myotonia in DM1 mice after intramuscular dosing, we conducted a comparative study on anti-myotonic against mexiletine, a drug with known anti-myotonia activity and also in development for DM1 [9], using the same previous experimental design (Figure 4b). In clinical practice, DM1 patients with myotonic disorders are treated with class I antiarrhythmics, with mexiletine being a commonly used and effective anti-myotonic drug [31]. We injected mexiletine or boldine intramuscularly into the left quadriceps of the hind leg, both at a concentration of 60 mg/kg (n = 4). We used the right quadriceps as the control by injecting the vehicle in which both compounds were dissolved (DMSO 20%). The animals were treated daily for three days, with electromyograms performed prior to the first injection and two days after the last dose. Before treatment, myotonia levels were high in all animals (dataconsistent with data shown in Figure 4a). Following treatment with mexiletine and boldine, the level of myotonia in the right quadriceps decreased in all the animals treated with boldine and in 75% of those treated with mexiletine. Myotonia quantification demonstrated boldine inducing a greater reduction in myotonia compared to mexiletine. To evaluate the duration of the anti-myotonic effect of both compounds, we performed the same experiment again, but now we continued with the evaluation of myotonia levels in the treated mice for 15 days (day of sacrifice once the mice recovered initial myotonia levels for both compounds). Our data also demonstrated a longer effect of boldine (myotonia levels reduced for >10 days) than mexiletine (myotonia levels reduced <10 days).

Based on all the above results and to further investigate the mechanism for boldine’s promising anti-myotonic activity, but potentially independent of the upstream dysregulation steps defined for DM1 pathogenesis, we tested its efficacy in a mouse model of inducible myotonia (Figure 5 and Appendix A). The compound anthracene-9-carboxylic acid (9-AC) can induce myotonia in wild-type mice by blocking the *Clcn1* channel, similar to that observed in the HSA^LR^ model, since myotonia is caused by the dysfunction of the same chloride channel [32]. After 9-AC administration (n = 7–9 mice), we measured the induced myotonia by quantifying the repositioning reflex time (TRR), i.e., the time it takes a mouse to roll over on all four legs after being placed in a supine position (mean of 0.5 s TTR before 9-AC treatment). Induction of myotonia (wild-type mice + 9-AC + DMSO) was established as a mean of >13 s TTR after 10 min of intraperitoneal 9-AC treatment. In this model of inducible myotonia, an anti-myotonic compound should reduce the TRR induced by 9-AC [33] (Appendix A). Intragastric administration of boldine (10 mg/kg) had a slight effect on reducing the TRR induced by 9-AC over time (30, 60, 90, 120, 180, and 240 min after 9-AC administration), indicating its potential ability to modulate myotonia levels from different chloride channel dysfunctions (Figure 5). For comparison, we again used mexiletine, with the same experimental conditions. This compound displayed a more robust myotonia reduction. Together, these results indicate that boldine is capable of modulating myotonia, but the activity is DM1-independent, and potentially linked to a more general mechanism of action. To consider the case of further boldine development, its comparison with mexiletine suggests a stronger effect when boldine is effectively delivered to the targeted tissue (intramuscular route), but a non-optimal drug-like profile instead, since systemic approaches displayed lower or null biological activities.

### 2.4. Study of the Mechanism of Action of Boldine

After the identification of different biological activities of boldine, we aimed to investigate the mechanism by which this natural compound would be mitigating DM1 phenotypes in the different disease models. Looking for a connection to the consistent focal reduction, we first evaluated if boldine has the ability to bind to the unusual RNA hairpins formed by long CUG repeats. To this end, we conducted gel retardation and polarized fluorescence assays able to detect molecule–molecule physical interactions. In the gel retardation assay, we labeled a 4-CUG RNA with carboxyfluorescein (FAM-CUG4), which is characterized as capable of in-vitro-forming the pathogenic double-stranded hairpin [34], and assessed whether or not boldine was able to interact with it. If boldine did indeed bind to the labeled RNA, it would have caused a decrease in the band quantification corresponding to the free RNA in the gel. However, we observed no significant decrease in the free RNA band quantification, as well as any band signal retention of in the top wells with increasing boldine concentrations, indicating that boldine does not bind to FAM-CUG4 (Figure 6a).

We confirmed this after utilizing a polarized fluorescence (FP) assay as a second approach to investigate the boldine affinity to CUG-expanded RNA (Figure 6b). The FP assay relies on the polarized (or not) light emission by an excited fluorophore, depending on the mobility of the fluorescent molecules in the well. Large molecules rotate slowly and emit polarized light, whereas small molecules emit depolarized light. Low levels of polarization indicate that fluorescent molecules move freely in solution (not binding), while high levels of polarization imply the presence of a large molecular complex (binding). By measuring changes in the polarization of an RNA consisting of 23 CUG repeats labeled with carboxyfluorescein (FAM-CUG23) [35], we investigated the potential binding between boldine and the RNA at various concentrations. We used pentamidine, a small molecule commonly used as an antifungal and capable of displacing MBNL1 because of binding to the repeats [34] as a positive control, and FAM -CUG23 in the presence of DMSO as a negative control. Although we observed a statistically significant increase in polarization in the presence of pentamidine, boldine did not. Together, both in vitro assay results indicate that boldine does not have the ability to bind to RNA formed by CUG repeats, and support the initial theory of a mechanism of action regardless of the close interaction with the toxic trait. Secondly, we investigated whether boldine’s effect on focal reduction was linked to its impact on the expression of MBNL1 or DMPK, most common therapeutic targets in drug development programs in DM1 [8]. We quantitatively assessed by qRT-PCR the expression levels of MBNL1 and DMPK in myoblasts derived from patients treated with 100 µM boldine or with DMSO (1%). The results showed no significant changes at this level (Appendix A). These findings point to a different mechanism of action of boldine on the focal dynamics, disregarding the direct activity on key DM1 factors.

Since the observed anti-myotonic activity of boldine is similar to the mexiletine drug, a sodium channel inhibitor, we hypothesized that its mechanism of action could also be linked to this ability. Therefore, we compared the modulating effect of both small molecules on Na^+^ ionic channels using chromaffin cells from the bovine adrenal gland as the experimental model (Appendix A). To determine if boldine and mexiletine affected inward currents through voltage-gated sodium channels (INa), we conducted electrophysiological studies using the conventional whole-cell patch-clamp technique in bovine chromaffin cells. The results showed that boldine caused a statistically significant blockade of INa at the highest concentration tested (100 µM), and mexiletine partially blocked the current, but without being statistically significant, at the same concentration. These findings suggest that both boldine and mexiletine produce a partial blockade of INa, with boldine having a superior impact compared to mexiletine in this assay. Together with the previous in vivo results, boldine’s anti-myotonic activities could come from a wide spectrum activity on the blocking of ion channels, similar to the mechanism of action defined for mexiletine.

Given the presented results, boldine displays significant, but not connected, positive activity on two DM1 hallmark features: focal formation and myotonia presence. The mechanism of action linked to significant focal reduction was not identified, but seems to work out the range of the most accepted DM1 disease targets, such as the CUG_exp_ or the MBNL1 factor. However, because of the validation across different disease models, this is still robust enough to suggest further studies to try elucidating its potential biological significance through the DM1 pathogenic mechanism. Likewise, an additional evaluation of boldine, to better characterize the broad anti-myotonic spectrum identified, is strongly recommended for validating its use in DM1 along with other disease-linked myotonia. Moreover, the exploration of the boldine’s drug-like profile seems a promising goal, since it is a natural compound with already known therapeutic properties potentially applicable to DM1 pathogenesis modulation (see the Discussion section).

## 3. Discussion

Boldine is the predominant and most characteristic constituent of boldo (*Peumus boldus*), a Chilean tree traditionally employed in folk medicine and recognized as an herbal remedy in multiple pharmacopoeias [16]. Several publications define this natural alkaloid as a small molecule with an expansive range of biological activities, including in muscle, which are promising for its development as a drug [15,16,17,18]. For our purposes, boldine was identified after an in vivo high-throughput screening using DM1 spliceosensor flies, as described in [19]. The primary activity of boldine, observed as increased luciferase levels due to the modulation of the *INSR* minigene from DM1 to the no-DM1 condition in the presence of CTG repeats, was validated from the quantification of independent spliceosensor flies’ (8 points test), and without detecting unspecific off-targeting effects (Appendix A). Therefore, the use of spliceosensor flies offers a valid approach for identifying candidate anti-DM1 molecules, yet without connecting activity to a specific mechanism of action. These results have led to a new line of research focused on validating the anti-DM1 activity of the hits, presenting results for boldine in the skeletal muscle via different DM1 disease models.

DM1 patients display aberrant appearance of ribonuclear foci in their cells, characteristic of diseases caused by toxic RNAs and displayed by many DM1 models, including flies expressing 480 CTG repeats [5,36]. Boldine significantly reduced the number of foci in muscle cells (Figure 2, Figure 3 and Appendix A) from the three disease models used herein (Drosophila, patient-derived cells and mice). While the pathogenicity of the foci is still controversial [5], focal reduction has been used to screen anti-DM1 drugs or disease modulators [37]. Importantly, our observations indicate that boldine acted in mammalians’ setting, ruling out any specific effect on Drosophila and confirming its potential to the likelihood of its beneficial effect in humans. However, in patient-derived myoblasts, boldine treatment did not change the distribution of MBNL1 in the cell nucleus, which is inconclusive as to its ability to reduce the above-mentioned focal formation, but supports treatments of the recovery characteristic DM1 splicing dysregulations (presence of fetal forms) of genes controlled by MBNL1, in either cells or in the HSA^LR^ mouse model. These results are backed by additional data. Boldine treatments did not impact the modification of MBNL1 or DMPK transcript levels in the same human cells (Appendix A). Neither did the compound display direct binding affinity to the “toxic” hairpin CUG–RNA structure (Figure 6), a hallmark of DM1 disease and one of the main therapeutic targets on exploration [8,38].

Our results suggest that despite the effect of boldine, a significant portion of MBNL1 remains trapped in the cell nucleus. Thus, a consistent reduction in the foci after boldine treatments could be supported by a still uncharacterized mechanism of action, different to the direct targeting of the CUGexp–MBNL1 duality (reviewed in [19]). Boldine’s mode of action might be connected to the natural dynamics of the formation of these aggregates, which is more puzzling than previously thought and could involve new-fangled proteins, such as DDX5 and DDX6 RNA helicases [5]. The alkaloid may act as a direct modulator of the focus homeostasis or regulate one of the proteins involved in this process, leading to a transition into a soluble state. If boldine was able to modulate focal solubility, MBNL1 release would be lower than in the total dissolution, therefore leading to a decreased number of aggregates detectable via fluorescent in situ hybridization without significant MBNL1 release detectable via immunohistochemistry. In vitro binding assays have shown that DDX5 increases the binding capacity of MBNL1 to RNAs carrying the repeats CUG, GAG, and CCUG, suggesting that DDX5 acts as a modulator in the binding of MBNL1 to toxic repeats. On the other hand, the decreased DDX6 in DM1 fibroblasts causes an increase in the intensity and frequency of the foci, suggesting that DDX6 could modulate the homeostasis of these aggregates in DM1 cells, favoring the passage toward a diffuse form. Learning about this connection is suggested as a line of further exploration regarding boldine intracellular mechanism of action. A biological effect of boldine due to DNA structure or replication interference, connected to its antioxidant and anti-proliferative properties, also cannot be ruled out as a result, since this molecule has been identified, for example, as a modulator of telomerase and topoisomerase I activities [39,40,41,42].

The more relevant results of our study show that boldine is capable of reducing the functional phenotypes of DM1, connected to premature aging and muscle dysfunction features. Patients with DM1 experience a higher risk of cancer and progressive dysfunction of various systems, hence its resemblance to a progeroid syndrome [43,44,45]. In DM1 model flies, continued treatment with boldine resulted in a marked improvement in longevity (Figure 2). In other studies, boldine has demonstrated significant transcription regulation of factors such as AKT, GSK3β, and IL6 [42,46], which are dysregulated and connected to muscle wasting in DM1 [47,48]. Overall, it is well documented that a multi-level mechanism of action, including a powerful antioxidant effect, underlies the beneficial activities of boldine. This multi-modal behavior has also been elucidated here after observing the separated activities on foci and myotonia phenotypes, initially hypothesized as connected. Boldine acts on different free radicals, significantly reducing reacting oxygen species (ROS), and influencing key factors for cells such as survival, differentiation, metabolism, and proliferation [49,50,51,52]. The increased levels of reactive oxygen species/free radicals and decreased antioxidant levels also seem to play an important role in the pathogenesis of DM1 [53]. Further studies into the way boldine normalizes all these factors and pathways in DM1 are clearly justified.

Finally, boldine treatments also consistently reduced myotonia in the HSA^LR^ mouse model (Figure 5), a symptom that causes rigidity in specific muscles after a voluntary contraction, resulting in a lack of dexterity, gait problems, difficulty speaking and swallowing, and muscle pain [1,4]. It is widely accepted that myotonia in DM1 stems from alterations in the splicing of the CLCN1 chloride channel, which leads to a reduction in functional protein, causing a decrease in chloride conductance and a depolarization in the membrane potentials of muscle fibers [49]. Boldine reduced myotonia in a dose-dependent manner. However, no improvement was observed on the exon 7a inclusion of the type 1 chlorine channel (*Clcn1*) splicing. Myotonia could contribute to generating other disease symptoms given that in DM1 patients’ excess of ions released during myotonic discharges aggravates fiber degeneration in muscles [50]. Boldine treatment may indirectly restore functional CLCN1 roles. Our preliminary observation for boldine interaction with sodium membrane channels suggests that it could be acting on ion fluxes involved in triggering myotonia (Appendix A). A further exploration of the modulation of different ion channels is recommended to shed light on how boldine lessens myotonia levels. The reduction in myotonia resembled mexiletine’s mechanism of action, a well-characterized anti-myotonic drug in the clinical evaluation of DM1 [8]. Our results suggest that boldine could be effective in treating DM1 myotonia, but also non-dystrophic myotonias. The mechanism of action of boldine and mexiletine may not be identical, with boldine being more favorable in a context where different cellular alteration characteristics of DM1 are reproduced.

Our comprehensive validation approaches of boldine in DM1 disease provide several substantial take-home messages. (i) Although boldine was identified in screening based on the modulation of the DM1 spliceosensor in Drosophila, the same type of modulation was not observed in the mammalian setting. Although functional conservation, from fly to human, has been established for RNA-binding factors, such as CELF, MBNL, and RBPOX, involved in DM1 and DM2 spliceopathy [51], the splicing machinery may not exhibit an identical behavior in flies as in mammals. The fly process may be more sensitive to the action of chemical compounds or the activity splicing regulation that are masked in a more complex system such as human cells. Nonetheless, these results suggest that boldine may not be effective in treating splicing alterations in DM1 patients. (ii) Boldine’s multimodal cellular and systemic mode of action positions this natural molecule (and its derivatives) as a very promising candidate for the treatment of myotonic dystrophies. Its antioxidant potential is widely correlated with oxidative stress in the ageing observed in DM1. However, its additional positive impact on the complex DM1 physiopathology should also be underscored, as boldine is also known to restore nitric oxide levels [54], upregulated in muscular dystrophies [55], and can also alleviate pain [15], an underestimated finding in DM1 [56]. (iii) Boldine activity has a solid effect on muscles, recently supported in studies performed in additional diseases [57], but also has important effects relevant to brain-connected diseases [46,58], with the potential to extend its therapeutic activities to the central nervous system (SNC) features also present in DM1 [59]. 

Overall, our results do no disregard boldine as a promising small molecule for drug development in DM1, either alone or as a complement to other type of therapies in development [8]. The therapeutic evidence accumulating on the natural compound, boldine, will drive the first-in-human clinical trials. The preclinical characterization of boldine, still lacking in terms of drug-like property features, will be important to finally define the true potential of boldine during the next steps of drug development, either in DM1 or any other disease in which the compound has already displayed promising activity.

## 4. Materials and Methods

### 4.1. Drosophila Stocks

The yw; +;UAS-i(CTG)480 1.1, w; UAS-INSR:Luc#6, and UAS-Luc stable-transformed flies were acquired as previously described [19,36]. MHC-Gal4 flies were a kind gift from Eric Olson (University of Texas, Southwestern Medical Center, TX, USA). Drosophila stocks were grown at 25 °C in standard fly food (the recipe can be found on the Bloomington website, http://flystocks.bio.indiana.edu, accessed on 22 May 2023). 

### 4.2. Drosophila-Based Experimental Approaches

8-point test: With the Biomek FXP pipetting robot, 5 μL of boldine was dispensed, with a final concentration of 12.5 μM, in a row of eight consecutive wells of a 96-well plate containing 250 µL Drosophila culture medium. Two replicated plates were prepared. The control (DMSO 0.25%) was dispensed in the first row of each plate. Each well was seeded with one L1 larva of genotype MHC-Gal4>UAS INSR:Luc#6; UAS-i(CTG)480, using a COPAS embryo dispenser. The plates were incubated at 25 °C for 2 weeks, after which they were kept frozen at −20 °C until the day of data reading. For the quantification of luciferase activity, each adult fly present in each of the wells was homogenized in 150 µL of 1X buffer from the Luciferase Assay Kit System (Promega; Madison, WI, USA). Of each homogenate, 50 µL was transferred to white 96-well plates (Nunc; Roskilde, Denmark), where 10 µL of the luciferase reagent was added per well using the Envision Multilabel Reader dispenser. Luciferase levels were measured using this same plate reader. The differences between the two groups were calculated using a two-tailed Student *t*-test and *p* = 0.05.Quantification of transcription levels for the luciferase reporter and UAS/Gal4 system: Sampling plates were prepared following the method described in the previous section. To study the effect of boldine on luciferase expression, 3 L1 larvae of the MHC Gal4;UAS-Luc genotype were seeded in each well, while for the effect of boldine on the system Gal4/UAS, 3 L1 larvae of the Mhc-Gal4>UAS-INSR:Luc#6 genotype were sown. The methodology described in [19] was followed to quantify the individuals and subsequently determine luciferase activity.Detection of ribonuclear foci in Drosophila: 500 µL of Drosophila culture medium was dispensed in glass vials of 5 mL capacity (Fisher; Tucson, AZ, USA) with the Biomek FXP pipetting robot. Boldine was dispensed at a final concentration of 12.5 µM in 3 vials, while DMSO was dispensed (0.25%) in another 3 vials. In each vial, 10 L1 larvae of the MHC Gal4>UAS-INSR:Luc#6;UAS-i(CTG)480 genotype were seeded using the COPAS embryo dispenser. The vials were incubated at 25 °C for 2 weeks, after which time they were frozen and kept at −20 °C until the day of data reading. The thoraxes of the adults to be analyzed were fixed overnight in paraformaldehyde at 4% in PBS at 4 °C. Subsequently, they were kept in a 30% sucrose solution in PBS for 2 days. After incubation time, the thoraxes were embedded in OCT, frozen in liquid nitrogen and maintained at a temperature of −80 °C until processing. For a further evaluation of the flies, 15 µm cross-sections were obtained with the Leica CM 1510S cryotome (Wetzlar, Germany). Fluorescence in situ hybridization (FISH) was then performed. The slides with the thorax sections were washed three times for 5 min each with 1× PBS. Fresh acetylation buffer was added to the slides (580 µL of 0.1 M triethalonamine and 125 µL of 0.25% (*v*/*v*) acetic anhydride in 50 mL of water). After 10 min of incubation, they were washed three times (each wash for 5 min) with 1× PBS and prehybridized for 30 min with the hybridization solution (10 mL deionized formamide, 12 µL 5 M NaCl, 400 µL 1 M Tris-HCl, pH = 8, and 20 µL 0.5 EDTA, pH = 8, 2 g Dextran sulfate, 400 µL Denhart’s 50× solution, 1 mL of herring sperm (10 mg/mL), and H_2_O, to a final volume of 20 mL). The Cy3-labelled probe 5′CAGCAGCAGCAGCAGCA3′-Cy3 (Merck, Darmstadt, Germany;) was used after heating it to 65 °C. Dissolved in hybridization buffer (1/100) was added to the slides for 5 min. The probe was allowed to hybridize at 37 °C overnight in a humid chamber, in darkness. The next day, the slides were washed with 2× SSC (2 washes of 15 min) and PBS (3 washes of 5 min duration), keeping the preparations at 32 °C. Finally, the slides were mounted with Vectastain and the images were taken using a Leica DM2500 optical microscope with a 40× objective. We analyzed the images of at least three different fields for each individual and a minimum of three individuals per group. The number of foci per cell was quantified using the Photoshop image analysis program (vCS4-11.0). Between groups differences were analyzed using a two-tailed *t*-student test and *p* = 0.05.Longevity test: 15 male newborns with the genotype MHC-Gal4>UAS-i(CTG)480 were placed in tubes containing DMSO (0.25%) or boldine (50 µM) dissolved in 3 mL of the nutrient medium Drosophila. As a positive control, 15 age-matched males from the reference strain were placed in tubes with DMSO (0.25%) dissolved in 3 mL of nutritional medium. Four replicates were performed per compound, with a total of 60 flies finally analyzed. The flies were transferred to a new tube with freshly prepared food and compound in each one every 2–3 days, and the number of dead individuals was quantified during the change. The results were represented by the Kaplan–Meier survival curves, which show the number of survivors according to the time in days. The curves of the compounds were compared using the Kaplan–Meier test with the GraphPad program.

### 4.3. Binding RNA Assays

Gel shift binding assay: To carry out the gel retardation assay, an aliquot of carboxyfluorescein (FAM)–CUG4 RNA (Metabion; Steinkirchen, Germany) was diluted to a concentration of 300 nM in binding buffer (25 mM Tris-HCL pH 7.5, 100 mM NaCl, 5 mM MgCl_2_, 50 µM, ZnCl_2_, and 10% glycerol) to a final volume of 10 µL. RNA was incubated with boldine (at concentrations of 0.03 mM, 0.5 mM, 4 mM and 8 mM) or DMSO (control) at 37 °C for 10 min. 5× loading buffer (20% sucrose, 100 mM Tris-Boric (TB) pH 8, 25 mM MgCl_2_, 0.1% bromophenol blue) and allowed to cool on ice. Subsequently, it was loaded on non-denaturing gels of 8% polyacrylamide without EDTA, previously subjected to a current of 100 V for 30 min. Electrophoresis was carried out in 1× TB supplemented with 5 mM MgCl_2_ and 50 µM ZnCl_2_, with a pH of 8.5 at a temperature of 4 °C in Mini-PROTEAN 3 cuvettes (Bio-Rad; Hercules, CA, USA) at 240 V for approximately 30 min. The images of the gels were taken with the Typhoon 9400 fluorescence scanner (GE Healthcare; Chicago, IL, USA), using emission filters BP 520 and SP 526. The intensity of the free RNA band was quantified using the Image J software (NIH). The EC50 value was calculated using linear non-regression analysis.Fluorescence polarization assay: FAM-CUG23 (synthetic RNA composed of 23 CUG repeats and conjugated in 5′ with the fluorophore carboxyfluorescein. Metabion) was heated at 70 °C for 10 min and allowed to cool slowly on the bench top. Next, the probe was incubated with boldine or 1% DMSO (control) in lysis buffer (25 mM Tris-HCl pH = 7.5, 100 mM NaCl, 5 mN MgCl_2_, 50 μM ZnCl_2_, 10% glycerol, and 0.05% Tween 20) for 20 min on ice and in the dark. To obtain a binding curve of boldine binding to FAM-CUG23 (6 nM), boldine was used at the concentrations of 0.1 mM, 0.01 nM, and 1 mM. As the positive control, the FAM-CUG23 probe (6 nM) was incubated with pentamidine (P0547-Sigma) at various concentrations (0.01 mM, 0.1 mM, 1 mM). All assays were carried out in black 96-well plates, with a total of four replicates per assay and at a final volume of 200 µL per well. The polarization measurements were obtained through an Envision plate reader (Envision 2104, Perkin Elmer; Waltham, MA, USA), using the excitation filter FP480 and emission filter FP535. The millipolarization (mP) values were calculated for each of the compounds following the formula (mP = 1000 × (S − G × P)/(S + G × P)), where S and P were the counts relative to the parallel (S) and perpendicular (P) planes of the arousal, and the G (grid factor) was an equipment-dependent factor. The results were normalized to the values of the negative control (DMSO). Between-group differences were analyzed using a two-tailed Student *t*-test and *p* = 0.05.

### 4.4. Mouse Strains

The FVB Harlan reference genotype provided by Harlan (www.harlan.com, accessed on 22 May 2023). The transgenic HSA^LR^ DM1 model, expressing 250 CTG repeats under the control of the human actin promoter (kindly provided by Dr. Thornton, University of Rochester, Rochester, NY, USA) [26]. The CD-1 reference genotype provided by Charles Rivers (www.criver.com, accessed on 22 May 2023).

### 4.5. Mouse Experimental Approaches

Treatment routes: In order to test the activity of small molecules in mice, and characterize some of their pharmacokinetic properties, the compounds were administered in different ways. For intramuscular administration, the compounds were injected into the quadricep muscles of the right and left hind legs, using 1 mL Hamilton syringes at a volume of 10 µL. For intraperitoneal administration, 50 µL of the compounds were injected into the right lower quadrant of the mouse abdomen, using a 24-gauge short-bevel needle and a 1.2 mL syringe. An oroesophageal tube was introduced into the animal’s esophagus (2–3 cm), and a 1.2 mL syringe was used for intragastric administration. The compounds were administered by trained staff of the SCIE animal facility (University of Valencia, Burjasot, Spain).Detection of ribonuclear foci in HSA^LR^ mice: The dissection of the quadricep muscles of both hind legs was carried out at 4 °C. Immediately after the animals were sacrificed, half of each quadricep muscle was placed in a cryotome cast, immersed in OCT, and stored at a temperature of −80 °C until further processing. During its processing, 6 µm sections were obtained with the Leica CM 1510S cryotome. Fluorescence in situ hybridization (FISH) was performed for 6 µm thick quadricep sections, fixed (73% ethanol, 25% acid acetic acid, and 2% formaldehyde) for 30 min at 4 °C and pre-hybridized for 10 min with the pre-hybridization buffer (30% formamide, 2× SSC) at room temperature. The hybridization was performed in a dark chamber at 37 °C for 2 h, using the buffer hybridization (30% formamide, 2× SSC, 0.02% bovine serum albumin, tRNA of yeast (1 mg/mL), and 2 mM sodium metavanadate) and 2 ng/µL of the probe marked Cy3-5′CAGCAGCAGCAGCAGCA3′-Cy3 (Sigma). After hybridization, the samples were washed with the pre-hybridization buffer for 30 min at 45 °C, followed by a second wash with SSC05X for 30 min at the same temperature. The samples were mounted with the Vectashield solution with DAPI. The number of foci per nucleus were quantified using a Leica DM2500 optical microscope with a 63× objective, whereby the number of foci present in three different fields was noted, in each of which at least 25 cells were counted. The number of foci obtained was divided by the number of the counted cells. The between-group differences were analyzed using a two-tailed Student *t*-test and *p* = 0.05.Evaluation of alternative splicing via semi-quantitative RT-PCR: The other half of each muscle quadricep was frozen in liquid nitrogen and stored at a temperature of −80 °C, after which RNA was extracted from the samples. Approximately 40 mg of the muscle was homogenized in 1 mL of TriReagent (Sigma). The homogenates were left for 5 min at room temperature before 200 µL of chloroform was added to each sample. The mixture was stirred and left for 5 min at room temperature. The tubes were then centrifuged for 15 min at 12,000× *g* and 4 °C. The aqueous phase was transferred to a new tube, where 500 µL of isopropanol and 3 µg of glycogen (GlicoBlue™ from Ambion; Austin, TX, USA) were added after 10 min at room temperature; the samples were centrifuged again for 10 min. The supernatant was removed, and the pellet was washed with 1 mL of 75% EtOH. After another centrifugation for 5 min at 7500× *g* and 4 °C, the supernatant was removed, and the pellet was dried at room temperature. Once dry, it was resuspended in RNase-free water, previously heated to 60 °C. The amount of RNA obtained was quantified by measuring its absorbance at 260 nm using a spectrophotometer (Eppendorf BioPhotometer; Hamburg, Germany). All RNA samples were diluted to a final concentration of 0.5 µg/µL and stored at −80 °C. The genomic DNA present in the extractions was removed by the digestion of 2 µg of RNA with DNase I (Fermentas; Waltham, MA, USA), at a total volume of 8 µL (RNA 0.5 µg/µL, 1 µL of 10× DNase buffer, 1.5 µL Dnase, and H_2_O, with a final volume of 8 µL). The digestion was performed at 37 °C for 30 min. DNase I was inactivated through the addition of 1 µL of 25 mM EDTA, followed by 10 min at 65 °C. For the mold for cDNA synthesis, 5 µL of the DNase I digestion was added to the 8 µL Mix 1 (1 µL 10 mM dNTPs, 1 µL hexamers (Invitrogen; Waltham, MA, USA), and 5 µL H_2_O). To denature the DNA, the tubes were left at 65 °C for 5 min. Immediately afterward, the tubes were placed on ice and 7 µL of Mix 2 (4 µL of buffer) was added (5× Superscript, 2 µL 0.1 M DTT, 1 µL RNase Inhibitor (Invitrogen)) to each. Finally, 1 µL of the enzyme Superscript TMII reverse transcriptase (2U, Invitrogen) was added. To carry out the retrotranscription reaction, the mixture was heated to 25 °C for 10 min in a thermocycler (Mastercycler Eppendorf), followed by 50 min at 42 °C and 15 min at 70 °C. For the controls, the same reactions were performed either without the RT enzyme in the reaction mix or without the RNA template. The cDNA obtained was stored at a temperature of −20 °C. For the detection of the *Serca*, *Clcn1,* and *Gadph* transcripts, 2 µL of the cDNA obtained in the previous section together with 0.25 µL of the GoTaq polymerase enzyme (Promega) was used as a template for the PCR reaction, to which 10 µL of Flexi buffer was added (GoTaq, 3 µL Mg^2+^, 1 µL dNTPS (10 mM), 1 µL forward primer (Serca e21F/Clce6F/Gadph F)) (10 mM), 1 µL of reverse primer (Serca e23R/Clc e8R/Gadph R)(10 mM), and DNase-free H_2_O, to a final volume of 50 µL. The following program was used for the amplification: 95 °C for 10 min, 25 or 27 cycles consisting of 30 s at 95 °C, 30 s at 58 °C, 1 min at 72 °C, and a last cycle of 5 min at 72 °C. The PCR products were resolved on 2.5% agarose gel. The expected amplicon in the case of the transcribed *Serca* was of two bands (an upper one of 300 bp and a lower one of 225 bp), while the expected amplicon for the *Clcn1* transcripts was of a higher band of 450 bp and 350 bp in the case of the inclusion of exon 7, and 350 bp if exon 7 was excluded. The intensity of the bands was quantified with the Image J program (NIH), normalizing the exon 22 inclusion percentages of *Serca* and exclusion of exon 7 of the *Clcn1* gene with the values of the *Gadph* gene. The between-group differences were analyzed using a two-tailed Student *t*-test and *p* = 0.05. The designed primers and amplification temperatures for the reactions are detailed in Appendix A.Quantification of myotonia levels via electromyography (EMG): All animals were immobilized with anesthesia during electromyograms. The measurements were obtained with a 30 G concentric needle electrode using the TECA TD-20 MK II EMG/EP electromyograph. Ten measurements were carried out in the quadriceps of both legs of each individual. Myotonic discharges were classified according to the following scale: 0, no myotonia; 1, myotonic discharges in under 50% of insertions; 2, myotonic discharges in over 50% of insertions; and 3, myotonic discharges in practically all the insertions (>90%) [26].Quantification of repositioning response time (TRR) in a mouse model of chemically induced myotonia in mouse: On each test day, we prepared a 2.4 g/L solution of anthracene-9-carboxylic acid (9-AC, Sigma) in water and 0.3% bicarbonate. The volume of the solution required to obtain a 60 mg/kg concentration of 9-AC, as previously published [33], was administered intraperitoneally to each CD-1 mouse included in this study. The compounds boldine and mexiletine (Sigma) were previously dissolved directly in saline and administered via an intragastric route at a concentration of 10 mg/kg. TRR was measured and quantified as the time taken for a mouse to roll over on all fours after being placed in a supine position. TRR was determined for each mouse at 10, 30, 60, 120, 180, and 240 min after the 9-AC administration, and calculated as the average of 10 measurements made at one-minute intervals. The between-group differences were calculated using a one-way ANOVA analysis of variance pathway, followed by an unpaired Student’s *t*-test.

### 4.6. Patient-Derived and Other Cell Lines

For experiments planned in DM1 cell models, the fibroblasts transdifferentiated into myoblasts through an inducible expression of MyoD were used. These fibroblasts were from patients with DM1 or healthy individuals (control), and were provided by Dr. López de Munain (Biodonostia Institute, Basque Country, Spain). The different lines used were: 9.56 with 1333 repetitions (DM1), 9.73 with 333 repetitions (DM1), 9.66 with 1000 repetitions (DM1), and 9.88 < 50 repetitions (non-DM1 control). All cell lines were grown in a medium of fibroblast proliferation, i.e., DMEM supplemented with 10% fetal bovine serum (FBS), penicillin (50 IU/mL), and streptomycin (50 µg/mL) (Invitrogen). The cells were maintained at 37 °C in a humidified incubator with 5% CO_2_. Chromaffin cells of the bovine adrenal gland were also used throughout this work for sodium channel evaluation (the cells kindly provided by the Teófilo Hernando Research Institute, Madrid, Spain).

### 4.7. Cell Experimental Approaches

Detection of ribonuclear foci: Fibroblast cells were seeded at a density of 10 cells/mL in 24-well plates on 12 mm circular coverslips with the growth medium. After 24 h, the fibroblast proliferation medium was replaced by means of transdifferentiation into myoblasts (DMEM supplemented with 2% of horse serum, penicillin (50 IU/mL), streptomycin (50 µg/mL), 100 µg/mL of Apotransferrin, 10 µg/mL insulin, and 2 µg/mL doxycycline), with a final volume of 500 µL per well. For the muscle cells to attach on the coverslip, they were left on the plate for 24 h. After this time, the compounds to be tested were added and left to act for another 24 h. Subsequently, the cells were fixed for 15 min with 4% PFA at room temperature room, washed 3 times with 1× PBS and stored in 70% EtOH at a temperature of 4 °C until the time of processing. The cells were treated with 100 µM boldine or 1% DMSO. Fluorescence in situ hybridization (FISH) was further performed. The cells were rehydrated with 1× PBS and pre-hybridized for 10 min at room temperature with the pre-hybridization buffer (40% formamide deionized, 2× SSC). Hybridization was performed in a dark chamber at 37 °C using the hybridization buffer (30% formamide, 2× SSC, 0.02% BSA, 1 mg/mL tRNA from yeast (Sigma), 2 mM sodium metavanadate, 1 μm/mL sperm DNA from denatured herring, 10% dextran sulfate, and 1 ng/µL Cy3-RNA probe 5′CAGCAGCAGCAGCAGCA3′-Cy3 (Sigma)). After hybridization, the cells were washed with the prehybridization buffer for 15 min at 45 °C (two washes) and once with 1× PBS at room temperature. The coverslips were mounted with the Vectashield solution with DAPI. The number of foci per cell nucleus were quantified in the Leica optical microscope using the 63× objective, quantifying the number of the foci present in three different fields, in each of which 25 cells were counted. The number of foci obtained was divided by the number of cells observed and analyzed using a two-tailed Student *t*-test and *p* = 0.05.Quantification of MBNL1 distribution via immunodetection: After the transdifferentiation process (see previous bullet), the compounds to be tested were added and left to act for 2 h. The cells were washed twice with 1× PBS and fixed for 15 min with 4% PFA at room temperature. Afterward, they were washed with 1× PBS for 5 min and stored in 1× PBS at 4 °C overnight. The cells were treated with 100 µM boldine or 1% DMSO. The whole process was carried out in a humid and dark chamber. A further immunohistochemistry approach involved the samples being washed with PBST (Triton × 0.3%) for 5 min (2 washes), after which 300 µL of blocking solution (1% donkey serum in PBST) was added for 30 min. After that time, the first antibody was added, i.e., 300 µL of anti-MBNL1 (Sigma) diluted in a ratio of 1:100 in the blocking solution. The antibody was left to incubate overnight at 4 °C. After three 5 min washes with PBST, the second antibody was added to each sample, which was 300 µL biotinylated anti-mouse (Fisher Scientific; Waltham, MA, USA) dissolved in the blocking solution in a ratio of 1:200, for 45 min at room temperature. Next, the samples were washed for 5 min with PBST (3 washes). For signal amplification, 300 µL was added to each sample of the Vectastain Elite ABC kit reagent AB (40 A:40 B:920 PBST ratio, prepared at least 30 min before use) and left to incubate for 45 min at room temperature. Subsequently, the samples were washed for 5 min with PBST (3 washes). Finally, 300 µL of Avidin Alexa Fluor 488 (Fisher Scientific) in a 1:200 PBST ratio was added to each sample and left to incubate for 45 min at room temperature. After 3 washes with PBS of a 5 min duration, the samples were mounted using Vestashield with DAPI. The images were taken by using an Olympus confocal microscope FluoView FV100 (SCIE), adjusting the blue channel conditions for DNA and blue channel green for MBNL1. The images were analyzed with the Image J program (NIH).Quantification of alternative splicing via semi-quantitative RT-PCR: The cells were seeded in 60 mm petri dishes at a density of 125,000 cells/plate, putting 3 mL of cells in each well. The fibroblasts were fixed for 24 h, and the fibroblast proliferation medium was changed by means of transdifferentiation into myoblasts. One day after the medium change, the compounds were added at the desired concentration and left to act for 24 h. Afterward, the cells were washed with 1× PBS and collected with a scraper. RNA was extracted from the cells collected in the previous section with the RNAGENTM Tissue Plus (Zygem; Solana Beach, CA, USA) kit, following the manufacturer’s instructions, using 8 µL of the RNA obtained as a template for reverse transcription. For the PCRs of cTNT, SERCA, and GAPDH, 1 µL of the cDNA obtained was used as template together with 10 µL Flexi Gotag buffer, 3 µL Mg^2+^, 1 µL dNTPs (10 mM), 1 µL forward primer (cTNT F/SERCA F/GADPH F) (10 mM), 1 µL of primer reverse (cTNT R/SERCA R/GADPH R) (10 mM), 0.25 µL GoTaq polymerase, and DNase-free H_2_O, to reach a final volume of 50 µL. The standard amplification conditions were used with an annealing temperature and a number of specific cycles for each gene (Appendix A). The PCR products were separated on a 2.5% agarose gel. The amplicon expected in all cases was two bands of different sizes: 244 bp and 201 bp in SERCA, and 132 bp and 110 bp in cTNT. The intensity of the bands was quantified with the Image J program (NIH) to obtain the percentages of inclusion of SERCA exon 22, and cTNR exon 5, normalized with the values of the GAPDH gene. The between-group differences were analyzed using a two-tailed Student *t*-test and *p* = 0.05. The designed primers and amplification temperatures for the reactions are detailed in Appendix A.Quantification of MBNL1 and DMPK expression levels via qRT-PCR: After the compounds were administered and the total RNA extracted, the DMPK F/DMPK R/MBNL1 F/MBNL1 R primers (Appendix A) were mixed with the cDNA samples and the Master Mix of the Power SYBR Green PCR kit (Applied Biosystems; Waltham, MA, USA), following the manufacturer’s instructions. The amplification in real time and subsequent quantification were carried out using Step One Plus (Applied Biosystems), following the manufacturer’s protocol. The designed primers and amplification temperatures for the reactions are detailed in Appendix A.Sodium ion channel electrophysiological study: Bovine chromaffin cells were isolated following a standard protocol [60], modified according to [61]. The cells were suspended in medium of Dulbecco MEM (DMEM), supplemented with 7.5% fetal bovine serum, 10 µM of arabinose cytosine, 10 µM fluorodeoxyuridine, 50 IU/mL penicillin, and 50 mg/mL streptomycin. For the ion current measurements, the cells were seeded in 24-well plates on 1 cm diameter circular coverslips, with a density of 105 cells per well. The inward currents into the cell through sodium channels (Ina) dependent on voltage were recorded using (whole cell) the patch clamp technique [62]. During the preparation of the cell membrane sealing process, the chamber contained a control Tyrode solution composed of 137 mM NaCl, 5.3 mM KCl, CaCl_2_, 2 mM, 1 mM MgCl_2_, and 10 mM HEPES pH 7.4. Once the membrane was broken, with the whole cell configuration of the patch clamp technique well established, the cell was rapidly perfused with an extracellular solution of a composition similar to the chamber solution, modified according to the current recorded: 0 mM Ca^2+^ for INa measurement. For recording the input ion currents, the cells were dialyzed with an intracellular solution formed of 100 mM CsCl, 14 mM EGTA, 20 mM TEA.CL, 10 NaCl mM, 5 mM Mg-ATP, 0.3 mM Na-GTP, and 20 mM HEPES/CsOH pH = 7.3. The current was recorded using an EP-10 amplifier (HEKA Electronic; Stuttgart, Germany) with 2–5 MΩ resistance electrodes. The data were acquired at a frequency between 5 and 10 kHz, and were later filtered to a frequency of 1 or 2 kHzData on weak currents (>25 pA), or resistances in a series > 20 Ω were recorded. Data analysis was carried out with HIEKA Elektronik and Igor Pro (Wavemetrics; Portland, OR, USA) PULSO programs.

## Figures and Tables

**Figure 1 ijms-24-09820-f001:**
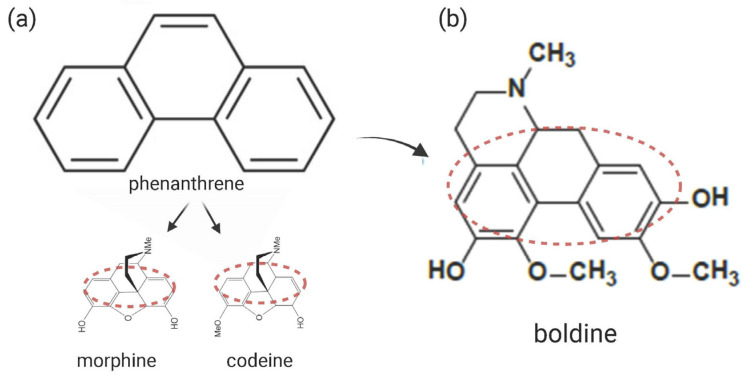
Phenanthrene-based molecules. (**a**) Common structure and presence of phenanthrene in alkaloid drugs (morphine, codeine). (**b**) Chemical structure of boldine. Similar regions are highlighted with a red dotted line. Created with BioRender.com (accessed on 24 May 2023).

**Figure 2 ijms-24-09820-f002:**
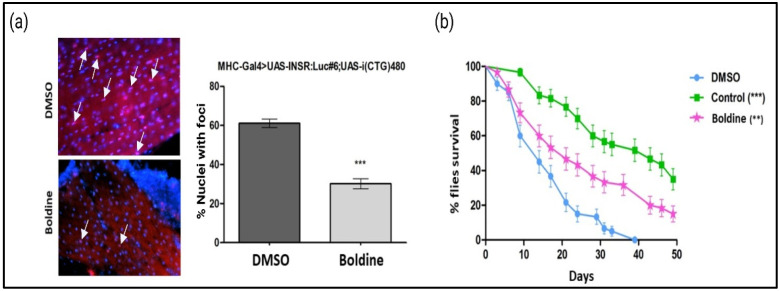
Evaluation of phenotypes after treatment with boldine in DM1 Drosophila models. (**a**) In situ hybridization of cross-sections of the musculature of flies treated with DMSO (0.25%) or boldine (12.5 µM). Cell nuclei were labelled in blue, while the CAG probe labelled the foci in red. White arrows indicate examples of merging red and blue signals. Boldine decreases the number of nuclei with foci. *** *p*-value < 0.001 calculated with an unpaired Student’s *t*-test using the program GraphPad Version 8.4.2. The error bars correspond to the standard error. (**b**) Longevity curves representing the percentage of the surviving flies as a function of time in days. Representation from the survival curves of the MHC-Gal4>UAS-i(CTG)480 flies (pink) treated with boldine (50 µM), or with (blue) DMSO (0.25%) and yw flies (green) treated with DMSO (control). *** *p* value < 0.001, ** *p*-value < 0.01 calculated with the Kaplan–Meier test using the GraphPad program. 40× optical microscope magnification. Created with BioRender.com (accessed on 24 May 2023).

**Figure 3 ijms-24-09820-f003:**
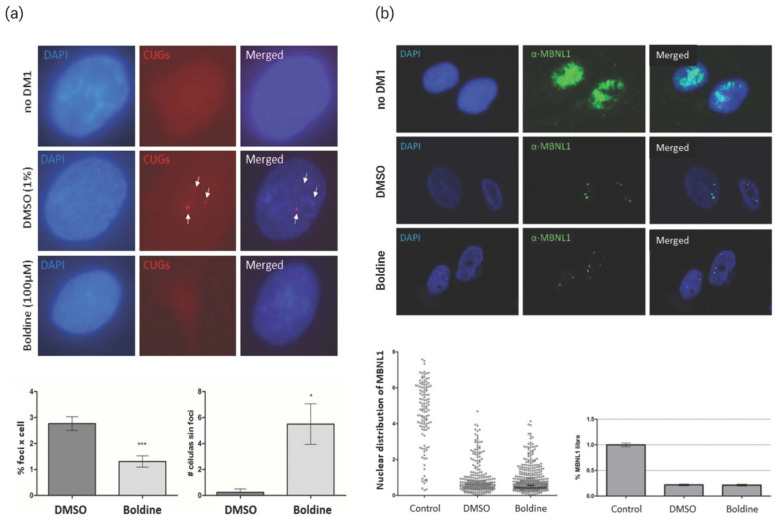
Evaluation of DM1 phenotypes after treatment with boldine in cells derived from patients. (**a**) Fluorescent in situ hybridization showing the ribonuclear foci in fibroblasts differentiated to myoblasts derived from patients treated for 24 h with 100 μM boldine or with DMSO (1%). Nuclei are shown marked with DAPI in blue, while the foci are shown marked by the CAG probe in red. No foci were found in the myoblasts of healthy individuals (no DM1). Examples of foci detection (white arrows) in DM1 cells without treatment. Quantification of the number of foci contained in the nucleus of patient-derived myoblasts treated with DMSO or boldine, *** *p*-value < 0.001. Quantification of the number of cells without foci after the treatment of patient-derived fibroblasts with DMSO or boldine * *p*-value < 0.01. Quantifications were carried out by counting at least 100 independent nuclei (25 nuclei in 2 fields) and four biological replicates. *p*-value calculated with an unpaired Student’s *t*-test using GrapPhad. The error bars correspond to the standard error. (**b**) Detection of MBNL1 through immunohistochemistry using an anti-MBNL1 antibody in myoblasts derived from patients treated for 24 h with boldine at 100 μM or with DMSO (1%) and in myoblasts derived from healthy individuals (no DM1). Nuclei are shown marked with DAPI in blue, while MBNL1 is shown labelled by a specific antibody in green. (**b**) Nuclear distribution of MBNL1 quantified as pixels present in the two channels (blue and green) per nucleus area, using confocal microscopy images and the Image J program (NIH). Percentage of free MBNL1 normalized with respect to the mean distribution of MBNL1 in non-DM1 cells. The study was carried out by counting at least 150 nuclei. No statistically significant differences were detected with Student’s *t*-test using the GrapPhad. The error bars correspond to the standard error. 63× optical microscope magnification. Created with BioRender.com (accessed on 24 May 2023).

**Figure 4 ijms-24-09820-f004:**
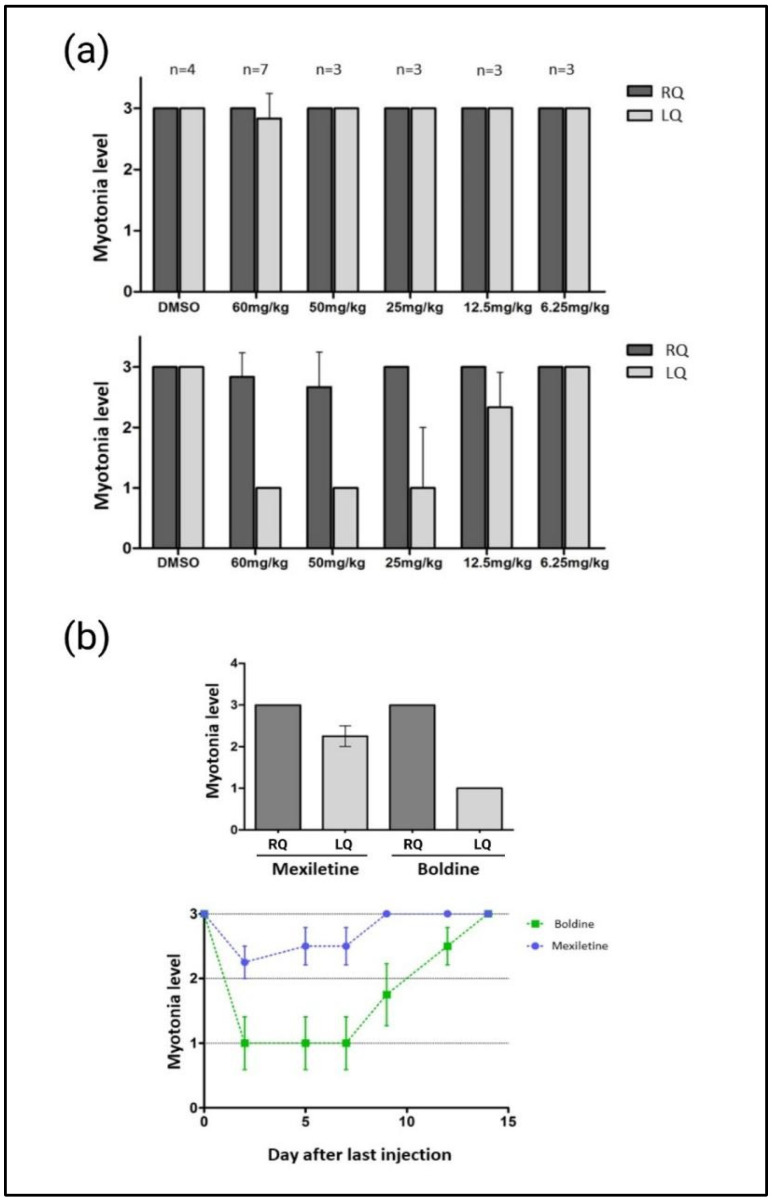
Intramuscular myotonia quantification after boldine and mexiletine in a HSA^LR^ mouse model. (**a**) To assess the efficacy of boldine, we injected different concentrations of boldine (60 mg/kg, 50 mg/kg, 25 mg/kg, 12.5 mg/kg, and 6.25 mg/kg) into the left quadriceps of the hind paw of mice between four and six weeks of age (n = 3 to 7). The right quadriceps served as an internal control and received DMSO (20%). Mice were treated with the compound or DMSO for three consecutive days and were sacrificed two days after the last dose. Myotonia was evaluated before treatment and the day of sacrifice. (**b**) Additionally, we compared boldine vs. mexiletine anti-myotonic activity. Top: myotonia levels in HSA^LR^ mice 2 days after intramuscular treatment with mexiletine (n = 4) or boldine (n = 4) at 60 mg/km in left quadriceps (LQ) or with the vehicle (DMSO 20%) in right quadriceps (RQ). The same experimental procedure in (**a**). Bottom: values of myotonia of the left quadriceps of HSA^LR^ mice treated with boldine or mexiletine measured for 15 days after treatment. The degree of myotonia is represented on a scale, where 0 means the absence of myotonia and 3 means the manifestation of myotonia in 90% or more of the times analyzed. The error bars represent the standard error. Created with GraphPad and BioRender.com (accessed on 24 May 2023).

**Figure 5 ijms-24-09820-f005:**
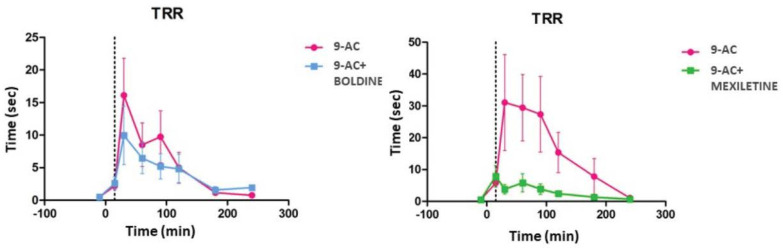
Evaluation of boldine and mexiletine activity after chemically induced myotonia. TRR values in mouse control at 30, 60, 90, 120, 180, and 240 min, after the injection of 9-AC (60 mg/kg) together with the DMSO vehicle (20%) (Pink curves), and in mice treated with 9-AC (intraperitoneal, 60 mg/kg) and boldine (intragastric, 10 mg/kg) (blue curve) or mexiletine (intragastric, 10 mg/kg) (green curve). For both, a TRR reduction was observed. Every curve shows the mean plus standard deviation of the TRR of 7–9 mice. Created with GraphPad and BioRender.com (accessed on 24 May 2023).

**Figure 6 ijms-24-09820-f006:**
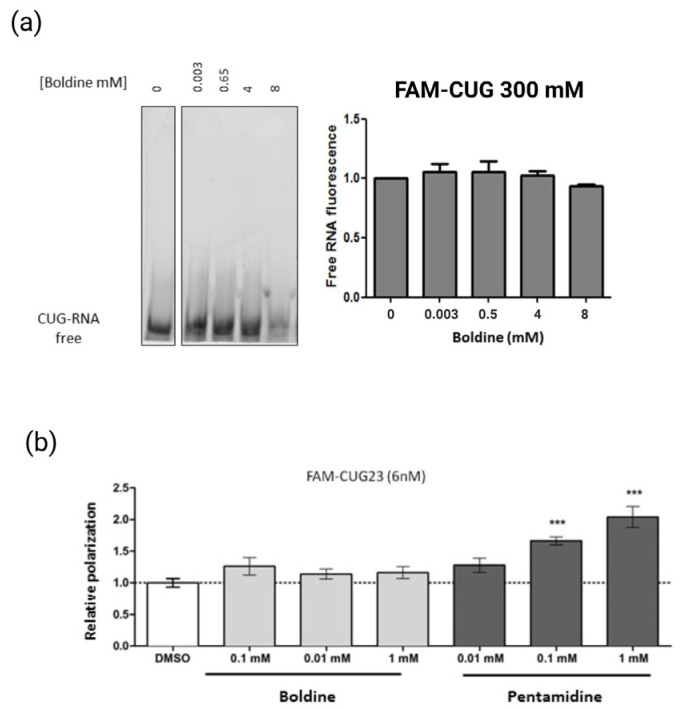
RNA–molecule binding assays. (**a**) Gel retardation of RNA formed by 4-CUG repeats labeled with FAM is not observed, either retained in the gel wells or after the quantification of free fluorescent RNA after an addition of boldine at different concentrations. The error bars correspond to the standard error. (**b**) Adding pentamidine (positive control) to the highest concentrations increased FAM CUG23 polarization caused by the union between both molecules. Adding boldine failed to result in an increase in FAM-CUG23 polarization compared to the negative control (DMSO) in any of the concentrations used. *** *p*-value < 0.001 calculated with an unpaired Student’s *t*-test using the program GraphPad. The error bars correspond to standard error. Created with BioRender.com (accessed on 24 May 2023).

## Data Availability

M.C.A.A. thesis online publication (2015): https://dialnet.unirioja.es/servlet/tesis?codigo=75953&info=resumen (accessed on 22 May 2023).

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
