# Peer review of "Natural Compound Boldine Lessens Myotonic Dystrophy Type 1 Phenotypes in DM1 Drosophila Models, Patient-Derived Cell Lines, and HSALR Mice"

_ijms, 2023, doi:10.3390/ijms24129820_

Round 1

Reviewer 1 Report

This study by Álvarez-Abril et al. describes the evaluation of boldine as a therapeutic for DM1. Boldine was identified based on structural similarity to the phenanthrene type of alkaloid compound identified from a previous in vivo Drosophila spliceosensor screen. Boldine was evaluated in the present study using various models of DM1 including patient fibroblasts and myoblasts, a DM1 Drosophila model and the HSALR transgenic mouse model. Curiously, boldine appeared to reduce ribonuclear foci in all models tested but without an expected reversal of mis-splicing of key DM1 transcripts. Boldine treatment did however rescue myotonia in the HSALR model thus providing some therapeutic consideration, however this was through an independent mechanism. While interesting, the authors have to address at least 1 major question before being considered for publication. 

Major points

1) While the reduction in foci appears to be consistent, there does not appear to be any other consistent expectation from reducing foci such as rescuing mis-splicing in the systems tested. This finding would suggest that MBNL1 reduction could be driving the reduced foci following boldine treatment. In fact, in figure S5A, this appears to be the case, albeit modestly. Because boldine reduced foci and rescued myotonia without rescuing clcn1 exon 7a splicing in the HSALR mouse, it is crucial to measure and report MBNL1 levels (RNA and protein) following boldine treatment in the HSALR mouse model.

2) The study does not provide adequate description of how the spliceosensor flies work. The author has to read the previous publication to understand the assay better. The authors should summarize the platform at the beginning of the results so the reader can easily make sense if the data.

English language editing is required in the text and figures. Some of the figure labels are in Spanish and 'CUG' is often replaced with 'GUG'.

Reviewer 2 Report

The manuscript “Natural compound boldine lessens Myotonic Dystrophy type 1 phenotypes in DM1 Drosophila models, patients-derived cell lines and the HSALR mouse” describes the identification of boldine as a new anti-myotonic compound. This finding is important for the DM1 as well as for other myotonic diseases. While the main result of the study is very interesting and might have therapeutic application, the manuscript presentation is diffused. Some statements in the paper need revision or confirmation by experiments.  Thus, the re-organization of the manuscript by the emphasis on the main result would strengthen the manuscript.

Concerns:

1. The authors state in the Abstract that boldine reduces the CUG foci in several DM1 models. It was expected that the disappearance of the CUG foci would lead to the reduction of the mutant DMPK mRNA. However, the Q-RT-PCR showed no changes of the levels of DMPK mRNA in the treated DM1 cells. The authors should clarify whether they measured total DMPK mRNA or mutant DMPK mRNA.  To conclude whether disassembly of CUG foci does not affect the mutant DMPK mRNA, RT-PCR allowing separation of the normal and mutant DMPK mRNAs or Northern blot analyses should be performed. The authors should explain these potential problems or perform experiments to clarify whether the boldine does not change the amounts of the mutant DMPK mRNA.

2. The lack of Clcn1 splicing changes in the treated HSA mice with the reduced myotonia can be verified by Western blot analysis to see if Clcn1 protein is reduced in the treated DM1 mice. The text should be revised to explain this controversy, or the experiments should be performed to strengthen the conclusions.

Other comments:

1. The text of the manuscript needs a rigorous revision including spelling check and use of complete sentences. The figure legends should correspond to the description in the text. The labeling in the bar graphs should be clarified. The FISH images are very weak or blurry.

2. The manuscript can be shortened by focusing on the main result, which is a reduction of myotonia, mediated by boldine. The reduction of CUG foci seems to be a secondary result with unclear outcomes since no MBNL1 release or MBNL1 related splicing changes were noticed. The questions related to the lack of reduction of the mutant DMPK mRNA and a lack of Clcn1 mis-splicing in the treated HSALR mice, in my opinion, remain unanswered. Because of this, the statements such as “boldine’s …activity is independent of the mechanism of DM1…” should be modified since this conclusion requires more studies.

Reviewer 3 Report

The authors try to describe the effects of boldine compound on several in vitro and in vivo models of DM1. Potentially, this paper could be a masterpiece in the field of DM1 therapy. In fact, boldine ameliorates quite significantly disease phenotype. However, I have a lot of concerns about the muanuscript understanding and its scientific soundness.

- An extensive revision of the English Language is needed as the manuscript is totally unreadable.

- I have not understood if a drug screening have been performed and how it has been performed. It is not clearly described.

- There are few recent references. Most of them are older than ten years.

- RNA-protein binding assay should be better described as the results are not so easy to understand.

- Boldine has no effect on mouse TRR as described by authors. It seems to be not significant as described. The effect of Mexiletine seems to be more significant. Please, put the asterisks on the standard deviation bars to indicate significance (FIGURE 5). Again, standard deviations of 9-AC and 9-AC+boldine overlap; thus, there is no significance.

- TRR measurement is in contrast with results showed in figure 4, where asterisks indicating significance should be added. 

- The quality and resolution of the fluorescence images must be improved. Signals seem to be very low.

- Figure 1 is in a wrong place inside the text and separated by its legend.

- Primers used to amplify alternative splicing isoforms should be indicated in the methods section or, alternatively, in supplementary materials.

- Authors must carefully check for appropriate experimental controls. Some experiments lack the appropriate control. For example, TRR experiment lacks at least one control (untreated sample or sample treated with vehicle).

The quality of the English Language is poor throughout the manuscript. I suggest contacting an English mother tongue to correct it. There are many spelling mistakes and most of the phrases should be changed.

Round 2

Reviewer 1 Report

The critiques have been adequately addressed.

Reviewer 3 Report

I have no more comments. Authors have responded to all my comments.